# Phytoconstituents and Ergosterol Biosynthesis-Targeting Antimicrobial Activity of Nutmeg (*Myristica fragans* Houtt.) against Phytopathogens

**DOI:** 10.3390/molecules29020471

**Published:** 2024-01-18

**Authors:** Adriana Cruz, Eva Sánchez-Hernández, Ana Teixeira, Rui Oliveira, Ana Cunha, Pablo Martín-Ramos

**Affiliations:** 1Department of Biology, School of Sciences, University of Minho, Campus de Gualtar, 4710-057 Braga, Portugal; cruzadriana73@gmail.com (A.C.); anaspereirateixeira@gmail.com (A.T.); ruipso@bio.uminho.pt (R.O.); accunha@bio.uminho.pt (A.C.); 2Centre of Molecular and Environmental Biology (CBMA), University of Minho, Campus de Gualtar, 4710-057 Braga, Portugal; 3Department of Agricultural and Forestry Engineering, ETSIIAA, University of Valladolid, Avenida de Madrid 44, 34004 Palencia, Spain; eva.sanchez.hernandez@uva.es; 4Centre for Research and Technology of Agro-Environmental and Biological Sciences (CITAB), Inov4Agro, University of Trás-os-Montes and Alto Douro (UTAD), Quinta de Prados, 5000-801 Vila Real, Portugal

**Keywords:** biopesticide, indole alkaloids, phenylpropenes, plant extract, *Myristica fragans*, *Fusarium culmorum*

## Abstract

In recent years, nutmeg (*Myristica fragans* Houtt.) has attracted considerable attention in the field of phytochemistry due to its diverse array of bioactive compounds. However, the potential application of nutmeg as a biorational for crop protection has been insufficiently explored. This study investigated the constituents of a nutmeg hydroethanolic extract via gas chromatography-mass spectrometry and vibrational spectroscopy. The research explored the extract’s activity against phytopathogenic fungi and oomycetes, elucidating its mechanism of action. The phytochemical profile revealed fatty acids (including tetradecanoic acid, 9-octadecenoic acid, *n*-hexadecanoic acid, dodecanoic acid, and octadecanoic acid), methoxyeugenol, and elemicin as the main constituents. Previously unreported phytochemicals included veratone, gelsevirine, and montanine. Significant radial growth inhibition of mycelia was observed against *Botrytis cinerea, Colletotrichum acutatum, Diplodia corticola, Phytophthora cinnamomi*, and especially against *Fusarium culmorum*. Mode of action investigation, involving *Saccharomyces cerevisiae* labeled positively with propidium iodide, and a mutant strain affected in ERG6, encoding sterol C-24 methyltransferase, suggested that the extract induces a necrotic type of death and targets ergosterol biosynthesis. The evidence presented underscores the potential of nutmeg as a source of new antimicrobial agents, showing particular promise against *F. culmorum*.

## 1. Introduction

Agrochemicals, including herbicides, fungicides, and other pesticides, play a crucial role in safeguarding crops against pests and diseases [1]. However, the agricultural sector confronts a substantial challenge in effectively controlling fungal diseases, which exert a significant impact on crop yields and product quality. Fungi often produce toxins capable of harming plants even at low concentrations, disrupting their normal physiological functions [2,3]. Symptoms associated with these diseases include wilting, growth inhibition, chlorosis, necrosis, and leaf spots [4,5]. Phytotoxins profoundly affect plant development by primarily targeting cellular components such as the cell membrane, mitochondria, and chloroplasts [2,6]. They exert their effects through various mechanisms, including the inhibition of protein and nucleic acid synthesis, resulting in physiological disturbances, cell death, and, in certain instances, the death of the entire plant [7]. In parallel, the excessive and improper use of conventional fungicides contributes to the emergence of resistance in phytopathogenic fungi, posing significant challenges in both agriculture and medicine. Crops become susceptible to diseases, and individuals with compromised immune systems may face severe infections caused by fungal species resistant to treatment [8,9]. Prolonged exposure to synthetic fungicides also gives rise to additional significant issues, such as ecological contamination in soil, sediments, and waterways, genetic alterations in non-target organisms, and disruptions in the endocrine systems of aquatic animals [10,11,12]. Consequently, there is a growing demand for eco-friendly and natural fungicides to address these challenges.

In recent years, numerous research endeavors have been undertaken to explore alternatives to artificial fungicides [13]. A promising alternative to traditional methods involves the utilization of plant extracts, offering several advantages, including biodegradability, diverse biological activities, complex chemical diversity, and varied mechanisms of action. This minimizes the likelihood of pathogens developing resistance [14,15].

*Myristica fragrans* Houtt., commonly known as nutmeg, belongs to the *Myristicaceae* family and is a tropical evergreen tree native to Indonesia. Nutmeg is derived from the desiccated core of the mature seed and holds a significant historical role in traditional Indian medicine for addressing various symptoms such as anxiety, nausea, diarrhea, cholera, stomach cramps, chronic vomiting, hemorrhoids, headaches, psychosis, fever, rheumatism, and paralysis [16]. The extensive culinary and medicinal applications of *M. fragrans* have garnered attention in recent decades [17]. *M. fragrans* encompasses a diverse array of phytochemicals, including lignans, neolignans, diphenylalkanes, phenylpropanoids, terpenoids, alkanes, fatty acids, and fatty acid esters, as well as minor constituents such as steroids, saponins, triterpenoids, and flavonoids. Reported phytochemicals include myristicin, safrole, eugenol (Figure 1), elemicin, isoelemicin, *trans*-isoeugenol, dehydrodiisoeugenol, α-pinene, β-pinene, sabinene, linalool, linoleic acid, geraniol, 4-terpineol, palmitic acid, virolane, and myticaganal A and C [18].

Nutmeg is of interest due to its diverse biological activities, exhibiting anti-inflammatory, antioxidant, antibacterial, antifungal, antiobesity, antidiabetic, anticancer, analgesic, chemopreventive, hepatoprotective, neuropharmacological, and cardioprotective effects [17,19,20]. Recent reviews by Ha et al. [17] and Barman et al. [18] cover ethnomedicines, phytochemicals, pharmacology, and toxicity of *M. fragrans*. Several studies have demonstrated the inhibitory effects of metabolites from *M. fragrans* against pathogens such as *Colletotrichum acutatum, Colletotrichum fragariae, Colletotrichum gloeosporioides* [21], and *Aspergillus fumigatus, Aspergillus niger*, and *Aspergillus flavus* [22]. Additionally, *M. fragrans* exhibits bactericidal activity against clinical pathogens, including oral pathogenic strains of *Streptococcus* spp. [23], *Pseudomonas aeruginosa, Staphylococcus aureus, Bacillus subtilis*, *Micrococcus luteus* [24], and *Candida tropicalis* [25]. While these findings are promising, further research is warranted to fully explore the potential applications of this plant across diverse scientific domains. Existing studies have primarily focused on the essential oils of *M. fragrans*, emphasizing the need for additional investigations into its various extracts to unlock additional possibilities for applications.

Hence, this study aims to investigate the antifungal/antioomycetal activity of a hydroethanolic extract of *M. fragrans* (NME) against a wide range of phytopathogenic organisms that negatively impact agriculture and the economy. The study seeks to identify the major chemical entities composing the extract and elucidate the action mechanism of the extract, utilizing *Saccharomyces cerevisiae* as a fungal model.

## 2. Results

### 2.1. Extract Characterization

#### 2.1.1. Vibrational Characterization Using ATR–FTIR

The ATR–FTIR spectrum (Appendix A) revealed fatty acids’ characteristic bands, displaying strong absorption at ~2950 and ~1690 cm^−1^, along with bands in the fingerprint region, below 1500 cm^−1^ (Table 1). Furthermore, bands coinciding with those reported for eugenol derivatives [26] were detected, namely at 1428 cm^−1^ (C=C aromatic stretching) and 1235 cm^−1^ (C–O–C vibrations), used to quantitatively determine isoeugenol concentration in fragrance aerosols. The peak at 1330 cm^−1^ (C–O stretching vibration of methoxy group) is suggestive of the presence of methoxyeugenol.

#### 2.1.2. Phytoconstituents Elucidation by GC–MS

Based on library comparisons, the main phytochemicals identified in the NME chromatogram (Table 2, Appendix A) were fatty acids (40.4%), namely tetradecanoic acid (myristic acid, 21.3%), 9-octadecenoic acid (oleic acid, 10%), *n*-hexadecanoic acid (palmitic acid, 4.8%), dodecanoic acid (lauric acid) and its ethyl ester (2.9%), and octadecanoic acid (stearic acid, 1.3%). Other constituents were methoxyeugenol, (*E*)-isoeugenol, and (*E*)-methyl isoeugenol (11.1%); elemicin and (*E*)-isoelemicin (6.7%); 1,4,6-trimethyl-2-azafluorene (5.2%); gelsevirine (2%); veratryl acetone (1.6%); and montanine (1.1%), shown in Figure 2.

### 2.2. In Vitro Antifungal/Antioomycetal NME Activity

To evaluate the potential antifungal and antioomycetal activities of NME, various phytopathogenic organisms affecting economically important crops, including *Botrytis cinerea, C. acutatum, Diplodia corticola, Fusarium culmorum, Phytophthora cactorum,* and *Phytophthora cinnamomi*, were selected for this study. As illustrated in Figure 3, a consistent trend of decreasing mycelial diameter and increasing growth inhibition was observed with rising NME concentrations for all tested organisms.

For *B. cinerea* (Figure 3a,b), the causative agent of grey mold [27], *D. corticola* (Figure 3e,f), responsible for the decline of cork oak forests in the Iberian Peninsula [28], and *F. culmorum* (Figure 3g,h), the causative agent of Fusarium stem and root rot and Fusarium head blight [29], incubation with 500 or 1000 µg·mL^−1^ NME significantly affected mycelial growth. Notably, the mycelium growth of *F. culmorum* was inhibited by over 45% with the highest concentration of the extract (Figure 3g).

Exposure of *C. acutatum*, the causative agent of anthracnose in cereals, fruits, legumes, and vegetables [30], to 1000 µg·mL^−1^ NME significantly decreased (*p* < 0.05) mycelial growth (Figure 3d). The percentage of growth inhibition remained constant at 25.45% up to day 9 at the highest concentration tested, and then the inhibition gradually dropped to ~10% until the end of the experiment (Figure 3c).

Regarding *P. cinnamomi*, which causes root rot in various plants such as *Castanea* sp., *Eucalyptus* sp., *Banksia* sp., *Quercus* sp., and avocado [31,32,33], there was a significant reduction in diameter after 6 days at all concentrations tested (250, 500, and 1000 µg·mL^−1^; Figure 3j). The inhibitory effect of NME remained constant at all concentrations over the period of the experiment, showing a clear dose-response trend. After 6 days of incubation, inhibitions of 9.26%, 21.75%, and 25.23% were observed (Figure 3i).

Among all the organisms tested, *P. cactorum* showed no significant reduction in diameter during the 24 days of incubation with NME.

### 2.3. NME Toxicity Mechanism

After assessing the antimicrobial activity against fungi and oomycetes, an investigative approach using *S. cerevisiae* as a model organism closely related to phytopathogenic fungi and oomycetes was implemented. This study focused on typical targets of well-known synthetic fungicides, such as the ergosterol biosynthesis pathway (targeted by azoles) and the cell wall (targeted by echinocandins).

Ergosterol, an exclusive sterol in fungal plasma membranes, was chosen as a target for investigation due to its importance in the fungal cell membrane. Mutant strains affected in genes encoding enzymes catalyzing the final five reactions of the fungal-specific part of the sterol biosynthesis pathway were utilized to explore the potential interaction of NME in ergosterol biosynthesis. The mutants included *erg6*, affected in sterol C-24 methyltransferase (which converts zymosterol into fecosterol); *erg2*, affected in sterol C-8 isomerase (which converts fecosterol into episterol); *erg3*, encoding sterol C-5 desaturase (which converts episterol into ergosta-5,7,24(28)-trienol); *erg5*, encoding sterol C-22 desaturase (which converts ergosta-5,7,24(28)-trienol into ergosta-5,7,22,24(28)-trienol); and *erg4*, encoding sterol C-24 reductase (which converts ergosta-5,7,22,24(28)-trienol into ergosterol).

Except for the *erg6* mutant (Figure 4a), ergosterol biosynthesis mutants exposed to 1000 µg·mL^−1^ NME displayed similar sensitivity to the wild type (Figure 4). After 30 min, viability in all these strains was completely or almost completely abolished (Figure 4b–e). Regarding *erg6*, NME did not affect cell proliferation; however, growth was slower than the wild type (Figure 4b). Taken together, these results suggest that NME targets *erg6* itself and/or metabolites downstream in the ergosterol biosynthetic pathway, which would be absent in this mutant.

Similarly to the ergosterol biosynthesis pathway, the cell wall integrity signaling pathway (CWI) was also investigated as a potential target for the cell wall using mutant strains. Strains affected in BCK1 and MKK1/MKK2, kinases of the signaling cascade, showed similar sensitivity to NME as the wild-type strain. Considering that mutants affected in CWI are impaired in cell wall remodeling and have increased sensitivity to cell wall stress (reviewed by Levin [34]), it is unlikely that NME targets this cellular structure.

To differentiate between necrosis and apoptosis, *S. cerevisiae* cells were treated with NME and plasma membrane integrity was assessed by permeability to propidium iodide (PI). Loss of plasma membrane integrity is a characteristic feature of necrotic cell death, wherein necrotic cells become red fluorescent due to the intracellular accumulation of PI. As expected, cells in the control group did not exhibit red fluorescence, while cells exposed to a temperature of 90 °C displayed fluorescence (Figure 5), confirming the ability of PI in this assay to distinguish between viable cells and those subjected to conditions inducing necrosis.

Significantly, exposure to 500 µg·mL^−1^ NME substantially increased PI-labeling to approximately 93% compared to the control group (Figure 5a,b; *p* ≤ 0.0001). These results unequivocally indicate that NME induces a necrotic type of cell death.

## 3. Discussion

### 3.1. On the Phytochemical Profile

Considering the limitations in the identification of some compounds in the extracts due to the subset of known organic compounds available in GC−MS databases, particularly those with Qual values below 80, a word of caution is warranted. It is essential to acknowledge that the identification of these compounds may have some value but could also be inaccurate.

Among the chemical species identified by library comparisons (Table 1), the primary fatty acid, myristic acid, is expected to make a significant contribution to the antimicrobial activity of the NME. This expectation is grounded in previous reports demonstrating its effective inhibition of biofilm and hyphal formation in *Candida albicans* [35]. Proteomic analysis further revealed its targeting of proteins involved in various virulence pathways, including ergosterol synthesis, sphingolipid metabolism, multidrug resistance, and oxidative stress [35]. Additionally, oleic acid, palmitic acid, lauric acid, and stearic acid have been recognized for their antimicrobial properties [36].

Concerning other NME constituents, methoxyeugenol (2,6-dimethoxy-4-allylphenol or 4-allyl-2,6-dimethoxy phenol) is a phenylpropene, a methoxyphenol, and a derivative of eugenol. It is present in toxic Japanese star anise [37,38] and nutmeg extract, but not in nutmeg essential oil [39]. Elemicin (3,4,5-trimethoxyallylbenzene) is also a phenylpropene. It has been previously documented in *Anemopsis californica* (Nutt.) Hook. and Arn., *Asarum celsum* F.Maek. ex Hatus. and Yamahata, *Asarum sieboldii* Miq.*, Petroselinum sativum* Hoffm. [40], and *Daucus carota* L. [41]. These two chemical species are also anticipated to contribute to the antimicrobial activity of the extract. This expectation is based on previous findings reporting strong activity against *Campylobacter jejuni* for eugenol derivatives and elemicin [41]. The study also revealed that (E)-methylisoeugenol and (E)-isoeugenol, featuring the propenyl substructure, exhibited higher activity than (E)-methyleugenol and eugenol itself, which have the allyl substructure. This observation suggests that two oxygenated functions are necessary to induce antimicrobial activity [41], a criterion met by methoxyeugenol and elemicin.

Veratone or veratryl acetone is an aromatic ketone that acts as a proton donor, capable of forming hydrogen bonds with enzymes and other molecules. It is a significant intermediate in the synthesis of medicines like ipoveratril hydrochloride, tetrahydropalmatine, and depressor methyldopa. While its specific antimicrobial activity has not been documented, there is extensive research on the antimicrobial activity of polyphenolic compounds, which are related to veratone.

Gelsevirine is an indole alkaloid found in *Gelsemium sempervirens* (L.) J.St.-Hil. and *Gelsemium elegans* (Gardner & Champion) Benth. Although there is no specific information on the antimicrobial activity of 19-hydroxydihydrogelsevirine or gelsevirine, closely related compounds such as gelsemine have been demonstrated to possess antibacterial effects [42]. Montanine or 20-hydroxy-voaluteine is another alkaloid. It has been discovered in *Lycoris squamigera* Maxim., *Hippeastrum vittatum* (L’Hér.) Herb., and *Rhodophiala bifida* (Herb.) Traub [43]. Concerning 1,4,6-trimethyl-2-azafluorene, the Qual value is low (64) and the assignment may be off. An alternative assignment would be to 4-methyl-acridone (an alkaloid usually found in the family *Rutaceae*). Antifungal activity may also be anticipated for these alkaloids [44].

Concerning the degree of agreement of the phytoconstituents identified by GC–MS in the *M. fragans* hydroethanolic extract with those previously reported in the literature [18], the GC–MS analysis of nutmeg alcoholic extract showed the presence of four main compounds: myristicin (64.5%), myristic acid (18.7%), terpinen-4-ol (8.8%), and methoxyeugenol (8.1%) [45]. Myristicin was not detected in this study, but other usually reported alkenylbenzenes (viz. elemicin and isoelemicin) were found instead. The presence of myristic acid has been reported in *M. fragans* oleoresins [46] and in extracts or essential oils of *Myristica malabarica* Lam. and other members of the family *Myristicaceae*, including *Horsfieldia irya* (Gaertn.) Warb., *Iryanthera juruensis* Warb., and *Virola bicuhyba* (Schott ex Spreng.) Warb. Actually, the myristic acid content might serve as an indicator of the age of ground nutmeg [47]. According to Barman et al. [18], it is a usual constituent of nutmeg and wild nutmeg butter, together with oleic, palmitic, lauric, and stearic acids, all of which were detected in the extract studied herein. On the other hand, safrole and virolane phenylpropanoids were not detected. Concerning hydrocarbon and oxygenated monoterpenes and sesquiterpenes (such as α- and β-pinene or sabinene) and terpene alcohols (linalool, geraniol, or 4-terpineol), only α-terpinolene and γ-terpinene monoterpene hydrocarbons and selinene sesquiterpene were found (in small amounts). Methoxyeugenol, *trans*-isoeugenol, and methyl isoeugenol have also been previously identified in *M. fragans* extracts [45] or essential oils [18]. Gelsevirine and montanine have not been previously reported in *M. fragans*, but other indole alkaloids have been described in *Myristicaceae* family members. To the best of the authors’ knowledge, no reports on veratone in *Myristicaceae* are available, but it is worth noting that is structurally similar to alkenylbenzenes (see Figure 2).

According to Sanford and Heinz [47], the observed variability in the *M. fragans* extract composition should not only be ascribed to the extraction technique, but to other numerous reasons, such as differences in cultivation practices, maturity at harvest, pre-shipping storage conditions, and genetic mutations.

### 3.2. On Nutmeg Extracts Antimicrobial Activity

The antifungal activity of nutmeg against plant pathogens, especially *Colletrotrichum* species has been investigated in several studies. Among the species tested, nutmeg methanolic extract demonstrated significant antifungal activity against *C. acutatum, C. fragariae,* and *C. gloesporioides* [21]. Further investigation led to the isolation of three compounds of nutmeg with antifungal potential namely, erythro-(7*R*,8*R*)-Δ(8′)-4,7-dihydroxy-3,3′,5′-trimethoxy-8-*O*-4′-neolignan, erythro-(7*R*,8*R*)-Δ8′-7-acetoxy-3,4,3′,5′-tetra-methoxy-8-*O*-4′-neolignan, and 5-hydroxy-eugenol. However, upon further in vitro testing, they were found to be inactive against the three *Colletotrichum* species [21]. In another work, three lignans, namely erythro-austrobailignan-6 (EA6), meso-dihydroguaiaretic acid (MDA), and nectandrin-B (NB), were isolated from the methanol extract of nutmeg seeds [48]. In vitro antimicrobial activity testing revealed that these lignans displayed different levels of antifungal activity against various plant pathogenic fungi, including *Colletotrichum* species. The half-maximal inhibitory concentration (IC_50_) values ranged from 49 to 92 μg·mL^−1^ against *Colletotrichum coccodes* and from 12 to 55 μg·mL^−1^ against *C. gloeosporioides* [48]. In vivo experiments showed that all three lignans effectively inhibited the development of rice blast, caused by *Magnaporthe grisea,* and wheat leaf rust, resulting from *Puccinia triticina* infection. Additionally, EA6 and NB were highly effective against the development of barley powdery mildew and tomato late blight, respectively [48]. To the best of our knowledge, this is the first study to report the inhibitory activity of nutmeg extract against the fungi *F. culmorum* and the oomycete *P. cinnamomi.*

In addition to its antifungal properties, nutmeg has also been reported to possess antibacterial activity against Gram-positive and Gram-negative bacteria [23]. Myristicin isolated from nutmeg seed chloroform extract exhibited minimum inhibitory concentration (MIC) values in the 0.6 to 1.25 mg·mL^−1^ range against several clinical pathogenic bacteria [49]. Additionally, compounds like (*E*)-methylisoeugenol, (*E*)-isoeugenol, and (*E*)-methyleugenol identified in nutmeg have also demonstrated antibacterial activity [41]. Furthermore, nutmeg essential oil showed insecticidal activity against *Musca domestica* and *Chrysomya albiceps,* probably due to the presence of compounds like β-pinene (26%), α-pinene (10.5%), sabinene (9.1%), and γ-terpinene (8.5%) [50].

### 3.3. Mechanism of Action

The notable difference in sensitivity to NME between the *erg6* mutant and the other ergosterol biosynthesis mutants tested, as well as the wild-type strain (Figure 4), strongly suggests that sterol C-24 methyltransferase (the enzyme encoded by ERG6) and/or downstream metabolites in the biosynthetic pathway might be the target of the extract’s toxicity. Components of NME could bind to the enzyme and trigger toxicity through an unknown mechanism. While the possibility of intermediary metabolites being targets of NME is plausible, it is not the sole mechanism, considering that the *erg6* mutant accumulates several sterol metabolites downstream of the Erg6 step due to the low specificity of enzymes encoded by Erg2, Erg3, and Erg5, which can use zymosterol as a substrate [51].

The striking similarity in the phenotype of resistance to fluconazole exhibited by the ERG6 mutation [52] with the *erg6* mutant strain’s response to NME (Figure 4b) did not go unnoticed. It is known that when Erg11 is inhibited by azoles, Erg6, Erg25-Erg26-Erg27, and Erg3 are activated, leading to the synthesis of the toxic compound 14α-methylergosta-8,24(28)-dienol [53]. However, if NME had a similar mechanism of action as azoles, the *erg3* mutant would display a similar resistance phenotype, which was not clearly observed (Figure 4d). Therefore, it is evident that NME targets ergosterol biosynthesis in an unknown and likely complex mechanism involving multiple targets.

Labeling with PI (Figure 5) clearly indicates that NME induces necrosis in yeast cells, suggesting a similar effect on phytopathogenic fungi. Comparable results have been reported in the literature for *Paeonia lactiflora* Pall. ethanol extract and Mexican propolis with the human opportunistic pathogenic yeast *C. albicans* [54,55]. Additionally, spoilage fungi *Penicillium roqueforti* and *A. niger* also exhibited PI labeling with star anise (*Illicium verum* Hook. fil.) ethanol extract. Although studies on membrane integrity and necrotic cell death in phytopathogenic fungi and yeast models are limited, our results align with reported studies, emphasizing that natural extracts might be toxic through the induction of necrosis.

## 4. Materials and Methods

### 4.1. Microbial Organisms, Media, and Growth Conditions

The fungi, namely *B. cinerea* (provided by Richard Breia and Hernâni Gerós, Centre of Molecular and Environmental Biology, CBMA, University of Minho, Braga, Portugal), *D. corticola* (provided by Ana Cristina Esteves, Centre for Environmental and Marine Studies, CESAM, University of Aveiro, Aveiro, Portugal), *C. acutatum* (provided by Pedro Talhinhas, School of Agriculture, University of Lisbon, Lisbon, Portugal), *F. culmorum* (CECT 20493, obtained from the Spanish Type Culture Collection; Valencia, Spain), along with the oomycetes *P. cinnamomi* (provided by Helena Machado of the National Institute for Agrarian and Veterinarian Research, INIAV, Lisbon, Portugal), and *P. cactorum* (CRD Prosp/59, provided by Regional Diagnostic Center of Aldearrubia—Junta de Castilla y León, Salamanca, Spain), were cultivated on potato-dextrose-agar (PDA; BioLife, Italiana S.r.l., Milan, Italy) medium at 25 °C in the dark. Once the organisms reached the periphery of the Petri dish, they were stored at 4 °C.

Yeast strains derived from *S. cerevisiae* BY4741 (MAT*a*; *his3∆1; leu2∆0; met15∆0; ura3∆0*) from Euroscarf were the ergosterol synthesis mutants *erg2* (BY4741; MAT*a*; *erg2∆::kanMX*), *erg3* (BY4741; MAT*a; erg3∆::kanMX*), *erg4* (BY4741; MAT*a; erg4∆::kanMX*), *erg5* (BY4741; MAT*a; erg5∆::kanMX*), and *erg6* (BY4741; MAT*a; erg6∆::loxP-kanMX-loxP*) provided by Marie Kodedová and Hana Sychrová from the Czech Academy of Sciences (Prague, Czech Republic). The cell wall defective *bck1* (BY4741; MAT*a; his3Δ1; leu2Δ0; met15Δ0; ura3Δ0; YJL095w::kanMX4*) and *mkk1/mkk2* (BY4741; MAT*a; his31; leu20; met150; ura3Δ0; mkk2::kanMX4; mkk1::LEU2*) strains were also included. The parental strain W303-1A (MAT*a; ade2-1; ura3-1; leu2-3,112; trp1-1; his3-11,15; can1-100*), as well as the derived mutant *yca1* (W303-1A; MAT*a; yca1::kanMX4*), were provided by Francesc Posas from Universitat Pompeu Fabra (Barcelona, Spain). The yeast strains were cultured on yeast extract-peptone-dextrose (YPD) solid medium [1% *w*/*v* yeast extract (Acros Organics, Waltham, MA, USA), 2% *w*/*v* peptone (Biolife); 2% *w*/*v* dextrose (Scharlab, Barcelona, Spain); 2% *w*/*v* agar (LabChem, Pittsburgh, PA, USA)] at 30 °C for 2 days and subsequently stored at 4 °C.

### 4.2. Plant Material and Extract Preparation

Nutmeg, sourced from the ground seeds, was initially obtained in powdered form from an organic products store with the ‘ES-ECO-020-CV’ organic certification code. Five grams were extracted with 30 mL of 80% (*v*/*v*) ethanol water solution, in a water bath at 60 °C for 30 min, protected from light. The extract was filtered, centrifuged (5000 rpm, 10 min; Eppendorf 5804R, Eppendorf, Hamburg, Germany), and concentrated using a rotary evaporator (40 °C, 100 rpm; BUCHI Rotavapor^®^ R-100, BÜCHI Labortechnik AG, Flawi, Switzerland). After lyophilization, NME was kept at −20 °C, in the dark, until further use.

### 4.3. Extract Characterization

A Nicolet iS50 Fourier-transform infrared (FTIR) spectrometer from Thermo Scientific (Waltham, MA, USA), equipped with an attenuated total reflectance (ATR) system, was used to measure the infrared vibrational spectrum (over the 400 to 4000 cm^−1^ range) of the freeze-dried NME and gain insight into the functional groups present in it. The spectrum was obtained by combining 64 scans. Regarding the elucidation of the extract phytoconstituents, it was conducted by gas chromatography–mass spectrometry (GC–MS) at the Research Support Services (SSTTI) of Universidad de Alicante (Alicante, Spain). NME was dissolved in HPLC–grade methanol to yield a 5 mg·mL^−1^ solution, which was filtered. The system comprised an Agilent Technologies gas chromatograph (7890A) connected to a quadrupole mass spectrometer (5975C). The chromatography conditions included an injection volume of 1 µL, injector temperature of 280 °C (in splitless mode), and an initial oven temperature of 60 °C for 2 min, followed by a ramp of 10 °C/min until a final temperature of 300 °C for 15 min. Ultrapure He at a flow rate of 1.0 mL min^−1^ was used as a carrier gas. Separation utilized an HP-5MS UI column (Agilent Technologies, Santa Clara, CA, USA) measuring 30 m in length, 0.250 mm in diameter, and with a 0.25 µm film thickness. For mass spectrometry, the m/z scan range extended from 30 to 550 umas. The electron impact source and quadrupole temperatures were set to 230 °C and 150 °C, respectively, with an ionization energy of 70 eV. For equipment calibration, test mixture for apolar capillary columns according to Grob (Supelco 86501) and PFTBA tuning standards were used. Identification of components relied on comparing their mass spectra and retention time with the National Institute of Standards and Technology (NIST) and Wiley databases.

### 4.4. In Vitro Screening of NME Antifungal/Antioomycetal Activity

The effect of NME on the mycelial growth of phytopathogenic fungi and oomycetes was assessed using the Poisoned Food Method [56]. NME was incorporated into PDA medium at concentrations of 250, 500, or 1000 µg·mL^−1^ and subsequently poured into Petri dishes. An 8 mm diameter disc of mycelium from the phytopathogenic agents (as described in Section 4.1) was positioned at the center of the prepared plates. The plates were then incubated at 25 °C, protected from light, and the colony diameter of the microorganisms was periodically measured until the mycelium in the negative control (80% *v*/*v* ethanol, equivalent to the volume of the highest extract concentration) reached the edges of the Petri dish. The inhibitory activity was calculated as the percentage growth inhibition relative to the negative control (0% inhibition) using Equation (1):(1)$$Inhibition\;(\%)=\frac{d_{c}-d_{NME}}{d_{c}}\times 100,$$
where *d_c_* is the diameter of the control mycelium and *d_NME_* is the diameter of the mycelium exposed to the extract.

### 4.5. Cell Viability Assays

*Saccharomyces cerevisiae* cells, as described in Section 4.1, during the exponential growth phase (OD_600_ of 0.4), were subjected to treatment with 250, 500, or 1000 µg·mL^−1^ of NME. As a negative control, 80% (*v*/*v*) ethanol, equivalent to the volume used for the highest NME concentration, was employed. Subsequent to treatment, the cells were incubated at 30 °C and 200 rpm. Aliquots of 100 μL were taken at 0, 30, 60, 90, and 120 min of treatment, serially diluted in sterile deionized water to 10^−4^, and then 40 µL aliquots were plated on YPD solid medium. The plates were incubated for 48 h at 30 °C, and the resulting colonies were counted. Cell viability was determined as a percentage of the colony-forming units (CFU), with time 0 min considered as 100%.

### 4.6. Membrane Integrity of S. cerevisiae by Fluorescence Microscopy

Exponentially growing cultures of *S. cerevisiae* BY4741 (OD_600_ of 0.8) were exposed to 500 µg·mL^−1^ of NME for 120 min. An aliquot was harvested, centrifuged (7100× *g*, 6 min), resuspended in 1× phosphate-buffered saline (PBS), and subjected to exposure to 5 µg·mL^−1^ PI (Invitrogen, Waltham, MA, USA) for 20 min in the dark and at room temperature. Subsequently, cells were examined by fluorescence microscopy using a Leica DM-5000B microscope (Leica Microsystems, Wetzlar, Germany). Images were captured with a Leica DCF350FX digital camera and processed using LAS AF Leica Microsystems software v. 4.0.11706. The percentage of PI-labeled cells was calculated based on the number of PI-positive cells out of a total of 200 to 300 cells per replicate under each condition.

### 4.7. Statistical Analysis

Statistical analysis and graphical representation were conducted using GraphPad Prism version 8.4.2 for Windows (GraphPad Software, San Diego, CA, USA). Three independent experiments were performed for viability assays with *S. cerevisiae*, growth inhibition assays with filamentous fungi and oomycetes, and the assessment of membrane integrity with PI through fluorescence microscopy. The results are presented as mean ± standard deviation (SD), except for the membrane integrity of *S. cerevisiae*, where mean ± standard error of the mean (SEM) was utilized.

Prior to applying one-way ANOVA and Dunnett’s test for multiple comparisons between treatment groups and the control, the normality test was performed, and the homogeneity of variances between groups was verified with the Bartlett test. For the analysis of the results obtained with viability assays, a *t*-test was also applied at the last time point. The level of significance (*p*-value) for each test is indicated in the figures using the following notation: ns (*p* > 0.05)—non-significant, * (0.01 < *p* ≤ 0.05)—significant, ** (0.001 < *p* ≤ 0.01)—very significant, *** (0.0001 < *p* ≤ 0.001)—highly significant, and **** (*p* ≤ 0.0001)—extremely significant.

## 5. Conclusions

The most abundant phytochemicals identified in the *M. fragans* hydroethanolic extract (NME) were fatty acids (40.4%), including myristic acid (21.3%), oleic acid (10%), palmitic acid (4.8%), lauric acid and its ethyl ester (2.9%), and stearic acid (1.3%). Other major constituents were methoxyeugenol, (*E*)-isoeugenol, and (*E*)-methyl isoeugenol (11.1%); elemicin and (*E*)-isoelemicin (6.7%). In vitro screening of the extract’s antifungal/antioomycetal properties evidenced strong activity against *B. cinerea, D. corticola, F. culmorum*, and *P. cinnamomi*; moderate activity against *C. acutatum*; and no significant activity against *P. cactorum*. Investigation of the extract’s toxicity mechanism using *S. cerevisiae* as a model organism suggested that NME targets Erg6 and/or metabolites downstream in the ergosterol biosynthetic pathway and ruled out that the extract targets the cell wall integrity signaling pathway. Labeling with PI indicated that the extract induces a necrotic type of cell death. The reported findings put forward the hydroethanolic extract from *M. fragrans* as a promising candidate for environmentally friendly antifungal and antioomycete agents, particularly effective against *F. culmorum*. Its diverse targets, particularly within the ergosterol biosynthetic pathway, reduce the likelihood of selecting resistant strains, and being a natural product, it is unlikely to accumulate in the environment.

## Figures and Tables

**Figure 1 molecules-29-00471-f001:**
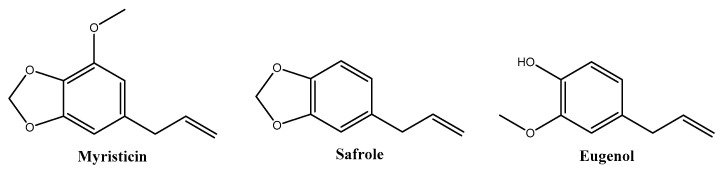
Major phytochemicals of *Myristica fragrans* extracts reported in the literature.

**Figure 2 molecules-29-00471-f002:**
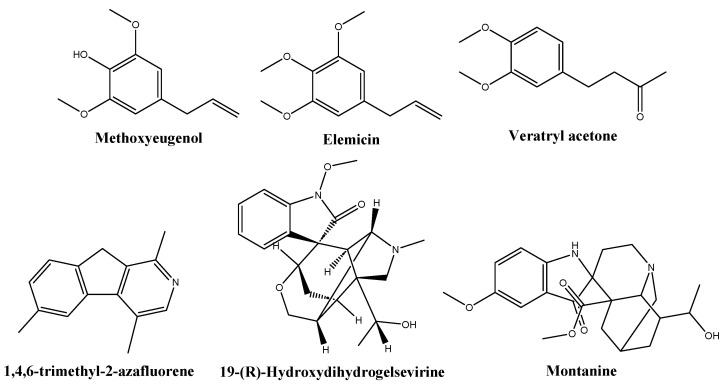
Structures of some of the phytochemicals identified in the *Myristica fragrans* hydroethanolic extract.

**Figure 3 molecules-29-00471-f003:**
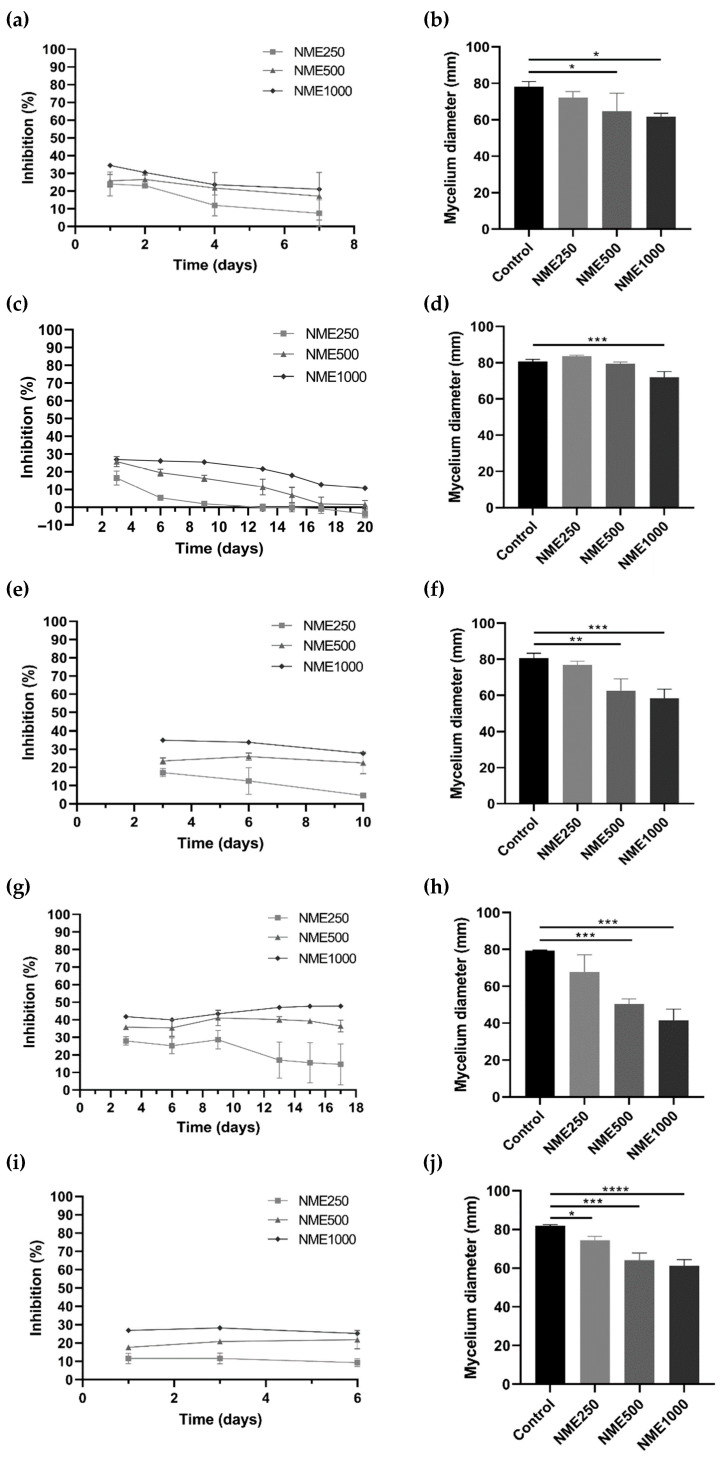
Mycelium growth inhibition (%) and mycelium diameter (mm) of (**a**,**b**) *Botrytis cinerea*, (**c**,**d**) *Colleototrichum acutatum*, (**e**,**f**) *Diplodia corticola*, (**g**,**h**) *Fusarium culmorum*, and (**i**,**j**) *Phytophthora cinnamomi* when exposed to *Myristica fragrans* ethanolic extract (NME) at 250 (NME250), 500 (NME500) or 1000 (NME1000) µg·mL^−1^ or 80% (*v*/*v*) ethanol as the control (same volume as NME1000). The mycelium diameter represents the diameter measured on the final day of the experiment for each tested organism: *B. cinerea,* 6 days; *C. acutatum*, 20 days; *F. culmorum*, 17 days; and *P. cinnamomi*, 6 days. The growth inhibition (%) was determined as the difference in the diameter of the mycelium of each treatment compared to the control mycelium. Results are expressed as mean ± SD of three independent experiments; one-way ANOVA, followed by the Dunnett test for multiple comparisons. * 0.01 < *p* ≤0.05, ** 0.001 < *p* ≤0.01, *** 0.0001 < *p* ≤0.001 and **** *p* ≤0.0001 indicate the significance level of the differences to the respective controls.

**Figure 4 molecules-29-00471-f004:**
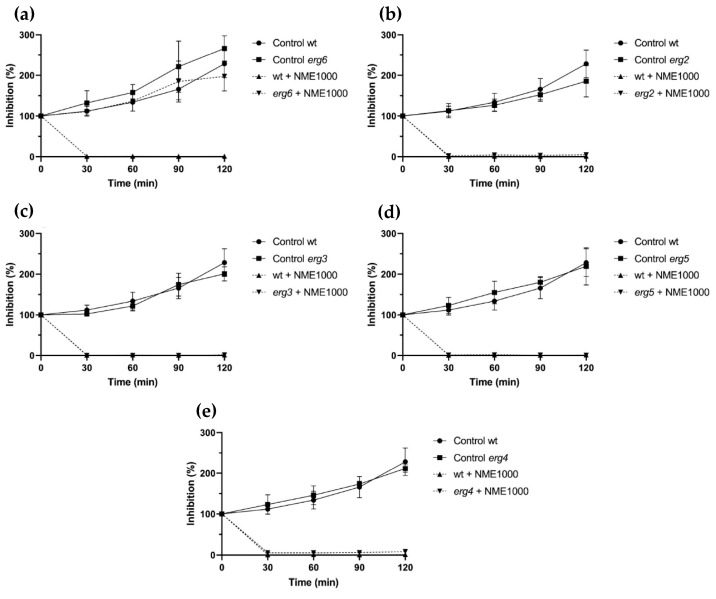
Viability of *Saccharomyces cerevisiae* BY4741 parental strain (wt) and the ergosterol biosynthesis pathway mutant strains (**a**) *erg6*, (**b**) *erg2*, (**c**) *erg3*, (**d**) *erg5*, and (**e**) *erg4* in the presence of 1000 µg·mL^−1^ *Myristica fragans* ethanolic extract (NME1000). Yeast cells were cultured in YPD at 30 °C and 200 rpm until the mid-exponential growth phase and the treatments were applied: 2% *v*/*v* solvent extract (80% ethanol; control) or NME1000. Aliquots were harvested over time, serially diluted up to 10^−4^, 40 μL drops were spread on YPDA plates, and colonies were counted after incubation for 48 h at 30 °C. Viability was calculated assuming 0 min as 100% viability. Data are presented as the mean of three independent experiments ± standard deviation.

**Figure 5 molecules-29-00471-f005:**
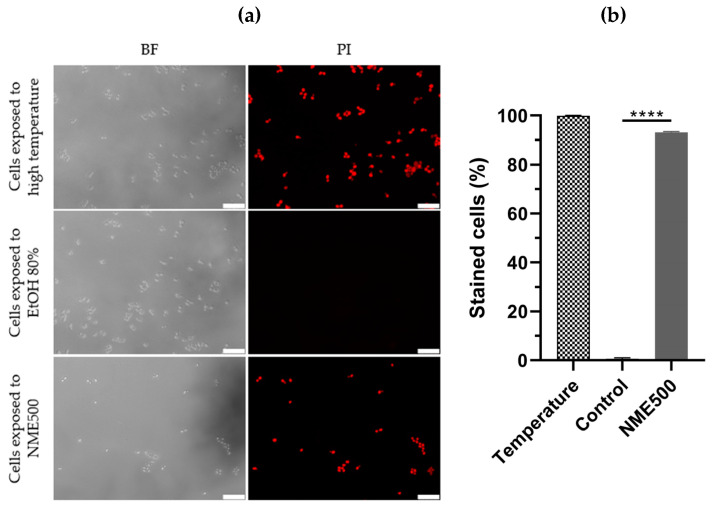
Plasma membrane integrity of *Saccharomyces cerevisiae* BY4741 cells exposed to 500 µg·mL^−1^ *Myristica fragrans* ethanolic extract (NME500). Yeast cells from mid-exponentially growing cultures in YPD at 30 °C and 200 rpm were treated with NME500 or 80% ethanol (EtOH; control; the same volume as NME500) for 2 h or exposed to 90 °C for 10 min (high temperature). An aliquot was harvested, centrifuged, and cells were suspended in 1× PBS with 5 µg·mL^−1^ propidium iodide (PI). Cells were observed under a fluorescence microscope in bright field (BF) and with a PI filter (**a**) and stained cells were calculated as the percentage of PI-positive cells in relation to the total count of cells (**b**). Data are presented as the mean of three independent experiments ± standard error of the mean (SEM). One-way ANOVA was conducted, followed by the Dunnett test for multiple comparisons: **** *p* ≤ 0.0001, with significance compared to the control at time 0. The cell images are representative of three independent experiments and were obtained with 400× magnification. The white bar corresponds to 25 µm.

**Table 1 molecules-29-00471-t001:** Main bands in the freeze-dried *Myristica fragrans* ethanolic extract infrared spectrum and their assignments.

Wavenumber (cm^−1^)	Assignment
2954	O–H stretching of most carboxylic acids
2915–2919	asymmetric C–H stretching of methyl and methylene groups
2848–2850	symmetric C–H stretching of methyl and methylene groups
1698–1704	carbonyl double bonds (C=O stretching)
1590	COO− stretching of the carboxyl group
1503	aromatic ring stretching
1455–1464	C–H bending
1423–1428	C=C aromatic stretching
1376	bending of methyl groups
1330	C–O stretching vibration of methoxy group (–OCH_3_)
1262–1285	unidentified ring mode
1236	C–O stretching; C–O–C vibrations
1212	CH_3_ symmetric deformation
1190	O–H bending
1125–1126	ring C–H bending; C–O–C
1036–1093	C–C stretching in the aliphatic chain
919–938	O–H out–of–plane bending
814–820	out–of–plane ring bending
720–726	–(CH_2_)*_n_*– in–phase rocking

**Table 2 molecules-29-00471-t002:** Chemical species identified by GC–MS in the *Myristica fragrans* ethanolic extract.

RT (Min)	Area (%)	Chemical Species	Qual
6.1021	0.3411	α-Terpinolene	97
6.7431	0.6730	γ-Terpinene	96
10.3577	0.4669	Benzene, 1-methoxy-4-pentyl- (or *p*-pentylanisole)	83
12.3104	3.3712	(*E*)-Isoeugenol	97
12.8506	0.8069	(*E*)-Methyl isoeugenol	98
13.2067	2.6466	1,3-Benzodioxole, 4-methoxy-6-(2-propenyl)-	98
13.5687	5.2184	Benzene, 1,2,3-trimethoxy-5-(2-propenyl)- (or elemicin)	98
13.6874	1.8965	Dodecanoic acid (or lauric acid)	99
13.8655	1.6491	3,4-Dimethoxyphenylacetone (or veratryl acetone)	96
14.1860	6.9547	Phenol, 2,6-dimethoxy-4-(2-propenyl)- (or methoxyeugenol)	98
14.2394	1.0242	3-(2-Methoxy-5-methylphenyl)propionic acid	58 *
14.6727	1.5085	(*E*)-Isoelemicin	98
15.4324	1.1836	Ethyl 2,2,5-trimethyl-3,4-nonadienoate	64
16.0912	21.3603	Tetradecanoic acid (or myristic acid)	99
16.1803	1.0401	Dodecanoic acid, ethyl ester (or ethyl laurate)	97
16.7857	0.9704	6b,7,8,9,10,10a-Hexahydrobenz[a]acenaphthylen-9-one	72
17.9490	4.7691	*n*-Hexadecanoic acid	99
19.6465	10.0477	9-Octadecenoic acid (*Z*)- (or oleic acid)	99
19.7770	1.3103	Octadecanoic acid (or stearic acid)	99
23.7181	0.7030	5-Keto-7-methyl-1,2,3,4,5,6,7,8-octahydro-2-quinolone	64
23.7834	0.8237	Phthalide, 4,6-dimethoxy-	50 *
23.8961	0.4049	9-Carbomethoxy-11-methoxy-6-hydroxy-5-oxoxantho[3,2-g]tetralin	64
23.9733	0.4450	2-(5-Hydroxy-1,6-dimethyl-4,11-dioxo-4,11-dihydro-1*H*-isochromano(7,6-f)indazol-8-yl)acetic acid methyl ester	55 *
24.0445	0.5452	*N*-(4-phenyl-1,2,5-thiadiazol-3-yl)-5-imino-4-phenyl-1,2,3-dithiazole 2-Oxide	90
24.2523	2.4660	Naphthalene, 1,4-dihydro-1-(diphenylmethylene)-5-hydroxy-4-oxo-	90
24.4541	2.1928	Methyl 3,5,7-trimethoxy-1-methylanthraquinone-2-carboxylate	90
24.4897	1.3533	19-Norpregna-1,3,5(10),17(20)-tetraene-20-carboxylic acid, 3-hydroxy-, methyl ester	95
24.5550	5.2001	1,4,6-Trimethyl-2-azafluorene	64
24.8399	0.4043	6α-methyl-5α-cholestane-3β,6β-diol	83
24.9408	2.1806	methyl (5*R*)-2-(methoxycarbonyl)-3-(dimethoxymalonyl)-5,9-dimethyldec-8-enoate	90
25.2494	0.8257	11β-Hydroxybenzo-18,20]pregna-4,20-dien-3-one	78
25.2909	1.1488	Voaluteine, 20-hydroxy-, (20*S*)- (or montanine (C22 alkaloid), or tabernaemontana)	92
25.3503	0.4605	Selinane	86
25.7124	1.9767	19-hydroxy-gelsevirine	94

RT = retention time; Qual = quality of resemblance. * Questionable assignments, with very low Qual values.

## Data Availability

The data supporting the findings of this study are available within the article and its Appendix A.

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
