# Peer review of "Phytoconstituents and Ergosterol Biosynthesis-Targeting Antimicrobial Activity of Nutmeg (Myristica fragans Houtt.) against Phytopathogens"

_molecules, 2024, doi:10.3390/molecules29020471_

Round 1

Reviewer 1 Report

Comments and Suggestions for Authors

This paper describes identification of antimicrobial substances in nutmeg and thier physiological activities. The findngs are novel. Although the mauscript is well-written, there are minor concerns that are needed to be improved. 

Figure 3: How was growth inhibition for each NME concentration calculated? Was it calculated as relative value (%) to 80% EtOH (control)? If so, it should be explained in the figure caption.

Figure 5: Scale bars should be provided for cell images.

Materials and Methods: Please provide the following GC-MS conditions. m/z scan range and type of carrier gas (He?) and its flow rate.

Comments on the Quality of English Language

There are typographical errors throughout the manuscript. The followings are some examples. In the materials and methods section, some of the symbol for dgree Celsius oC are underlining.  The en dash, but not hyphen, should be used for page ranges in each cited reference.

Reviewer 2 Report

Comments and Suggestions for Authors

Please, refer to PDF file

Round 2

Reviewer 2 Report

Comments and Suggestions for Authors

Please, refer to PDF file

Author Response

Dear Authors,

P3/L116-117: Please, replace all "(f)-"phenols with "(E)-"phenols in the text: Other constituents were methoxyeugenol, (f)-isoeugenol, and (f)-methyl isoeugenol (11.1%); elemicin and (f)-isoelemicin (6.7%);

Response: Corrected.

P4/Table 4: Please, replace (f)-Isoeugenol to (E)-Isoeugenol; (f)-Methyl isoeugenol to (E)-Methyl isoeugenol; (f)-Isoelemicin to (E)-Isoelemicin.

Response: Corrected.

P14/L477-478: Please, replace (f)-isoeugenol, and (f)-methyl isoeugenol (11.1%); elemicin and (f)-isoelemicin (6.7%) to (E)-isoeugenol, and (E)-methyl isoeugenol (11.1%); elemicin and (E)-isoelemicin (6.7%)

Response: Corrected.